# Method for quantifying the *Pasteurella multocida* antigen adsorbed on aluminum hydroxide adjuvant in swine atrophic rhinitis vaccine

**Masaya Yoshimura**\*, **Youko Honda, Emi Yonemitsu, Kasumi Kuraoka, Kiyotaka Suenaga**

Research & Development Department, Research & Development Section, Meiji Animal Health Co., Ltd., Kumamoto, Japan

\* yoshimura-ma@vet.meiji.com

**Data Availability Statement:** All relevant data are within the paper and its Supporting Information files.

## Abstract

Swine atrophic rhinitis is a disease caused by *Pasteurella multocida* and *Bordetella bronchiseptica* that affects pigs. Inactivated vaccines containing the toxins produced by *Pasteurella multocida* and *Bordetella bronchiseptica* have been widely used for the prevention of swine atrophic rhinitis. The efficacy of a vaccine is correlated with the amount of antigen present; however, the protective toxin of *P. multocida* bound to aluminum hydroxide, which is used as an adjuvant, can hinder the monitoring of the antigen concentration in the vaccine. This study assessed the applicability of a dot immunoassay as an antigen quantification method using monoclonal antibodies. This quantification method was able to detect the antigen with high specificity and sensitivity even when the antigen was bound to the adjuvant, and its application to vaccine products revealed a correlation between the amount of antigen present in the vaccine and the neutralizing antibody titers induced in pigs. The antigen quantification method presented in this study is a simple and sensitive assay capable of quantifying the amount of antigen present in a vaccine that can be used as an alternative quality control measure.

## Introduction

Swine atrophic rhinitis (AR), a common respiratory disease affecting pigs worldwide, is characterized by hypoplasia or atrophy of the nasal turbinates due to upper respiratory tract infections caused by *Pasteurella multocida* and *Bordetella bronchiseptica*. AR is rarely fatal; however, it is a threat to the swine industry as it is easily transmitted. Moreover, it impedes weight gain in affected pigs and reduces the production efficiency, resulting in economic losses. AR cannot be cured after it has progressed to the lesion stage. Therefore, formulation of preventive measures that combine improvement of the environmental conditions of the farm and a farm-based vaccination program is essential [1, 2].

The development of the upper respiratory lesions is likely triggered by infection with *B. bronchiseptica* or poor environmental conditions. *B. bronchiseptica* colonizes the nasal

**Funding:** The authors received no specific funding for this work.

**Competing interests:** The authors have declared that no competing interests exist.

passages of pigs and induces nasal mucosal inflammation. In contrast, *P. multocida* cannot colonize normal nasal mucosa; however, it can colonize nasal mucosa that is damaged by primary factors [3, 4]. The toxins produced by *B. bronchiseptica* and *P. multocida* cause AR. In particular, *P. multocida* produces a toxin that causes progressive nasal turbinate atrophy [3, 5]. The onset of AR is marked by the manifestation of acute catarrhal rhinitis symptoms, such as sneezing and a runny nose. Inflammation of the nasolacrimal duct is commonly observed following infection with *P. multocida*, and it can result in ductal occlusion and subsequent lacrimation of the medial canthus. Facial deformities due to the delayed development of the nasal, maxillary, and frontal bones usually become apparent one month after the onset of the disease. These clinical manifestations have been attributed to the mitogenic dermonecrotic toxins of *P. multocida* (PMT) [6]. Several vaccines have been developed for the prevention of AR. An attenuated *P. multocida* mutant expressing only the N-terminally truncated fragment of PMT that can increase antibody production has been used to prevent AR in pigs [7]. Vaccines produced using recombinant derivatives of PMT alone have also shown protective effects [8, 9]. Two bivalent and trivalent AR vaccines, Suimmugen rART$_2$ and Suimmugen rART$_2$/ER (Meiji Animal Health Co., Ltd., Japan), containing recombinant proteins expressed in *Escherichia coli* and aluminum hydroxide as an adjuvant have been available in Japan. Two antigens are present in these vaccines: the non-toxic derivatives of PMT and *B. bronchiseptica* dermonecrotic toxin. In addition, Suimmugen rART$_2$/ER contains the surface-protective antigen type A of *Erysipelothrix rhusiopathiae* to prevent swine erysipelas. As described by the approved application forms, these products can induce neutralizing antibodies and curb the action of PMT in vaccinated pigs.

Assays to evaluate the efficacy of these vaccine products are limited to in vivo testing, such as the immunization of pigs and mice. A previous study optimized the protein content of PMT in the vaccine dose to induce a protective antibody titer. Although the amount of PMT present in these vaccines is estimated using a physicochemical method during the vaccine manufacturing process, the amount of PMT bound to aluminum hydroxide, used as an adjuvant in the vaccine, remains uncertain. Therefore, the present study used dot blotting and immunostaining assays to determine the amount of PMT-immunogenic antigens present in vaccine products.

## Materials and methods

### PMT

The samples were prepared according to the method described in the veterinary drug approval documents for Suimmugen rART$_2$ and Suimmugen rART$_2$/ER. The PMT protein is expressed in the *E. coli* PRX-1 strain transfected with the plasmid pPRX-1 encoding the non-toxic PMT gene, *toxA-SQ*, of *P. multocida*, a toxin-producing pathogen. PMT was purified from the solubilized total protein via ion exchange chromatography using diethylaminoethyl -dextran and an elution buffer containing 0.5 M NaCl in 0.05 M phosphate buffer (pH 7.2) from the solubilized total protein of *E. coli*. The purified proteins were separated on a 5–20% w/v acrylamide gel using sodium dodecyl-sulfate polyacrylamide gel electrophoresis (SDS-PAGE), and their concentrations were compared with a bovine serum albumin standard using the GelDoc XR system (BIO-RAD; California 94547, USA) after staining with Coomassie blue R250.

### Adsorption of PMT on aluminum hydroxide

The samples were prepared according to the method described in the veterinary drug approval documents for Suimmugen rART$_2$ and Suimmugen rART$_2$/ER. PMT and aluminum hydroxide (ALHYDROGEL 85, CRODA Health Care. Shiga, Japan) were diluted with the elution

buffer to attain different concentrations. Subsequently, the PMT solution was mixed with aluminum hydroxide and incubated at 4˚C overnight with static. Thereafter, the mixture was fixed with 10% formalin (FUJIFILM Wako Pure Chemical Corporation.; Osaka, Japan), N-acetyl-DL-tryptophan (Ajinomoto Healthy Supply Co., Inc.; Tokyo, Japan), and L-lysine hydrochloride (FUJIFILM Wako Pure Chemical Corporation.; Osaka, Japan) at 4˚C overnight with static. Lastly, the aluminum hydroxide combined with PMT was washed with phosphate-buffered saline (PBS) and defined as PMT-alum. The samples were collected as a mixture of PMT and aluminum hydroxide supernatants before fixation and used for the detection of PMT that was not adsorbed on aluminum hydroxide.

### Anti-PMT monoclonal antibody (MAb)

The antibody-producing hybridoma cell line, 1F3, was obtained (provided by Prof. Horiguchi of Osaka University). Murine ascites from BALB/c mice with a hybridoma transplant possess the ability to neutralize PMT ($\geqq$256); consequently, they have been used as a monoclonal antibody (Mab). The MAb immunoglobulin was purified via affinity chromatography using Protein G, followed by ammonium sulfate precipitation. Thereafter, the purified MAb were labeled with horseradish peroxidase (HRP) using a Peroxidase Labeling Kit-NH2 (Dojindo Laboratories; Kumamoto, Japan).

### SDS-PAGE

All samples and standard BSA were mixed with 2 x Loading Buffer (0.1 M Tris-HCl (pH 6.8), 0.2 M DTT, 4% SDS), and boiled for 5 min. These samples and marker were applied to the gel (e-PAGEL, ATTO; Tokyo, Japan) and electrophoresed. After electrophoresis, the gel was stained with Quick-CBB PLUS (FUJIFILM Wako Pure Chemical Corporation.; Osaka, Japan) and then destained. The color intensities of the bands were scanned using an image analyzer (GelDoc XR; Bio-Rad), and images were captured.

### Desorption PMT from aluminum hydroxide

PMT-alum was centrifuged at 14,000g for 10 minutes and the supernatant was removed. After thoroughly suspending the pellet in 0.3 M phosphate buffer (pH 7.6), it was centrifuged at 14,000g for 10 minutes and the supernatant was collected. This step was repeated three times, and the supernatants were pooled. Two times the amount of cold acetone was added to the supernatant and suspended, and the mixture was reacted with static at -20˚C overnight. It was then centrifuged at 14,000g for 15 minutes and the precipitate was collected with PBS in the same volume as the original PMT-alum.

### Dot blotting and immunostaining assay

All samples were diluted with PBS containing 0.12 M NaCl in 0.05 M phosphate buffer (pH 6.8) before the assay. Subsequently, 200 μL of each sample was immobilized on a nitrocellulose membrane (Premium NC, 0.45; Amersham) using a suction blotter (Bio-Rad Laboratories). The membranes were soaked in 0.05% Tween 20 in PBS (T/PBS) containing 5% skim milk at room temperature (RT) for 1 h. After blocking the empty surface of the membrane, it was immersed in a 1:5000 dilution MAb in T/PBS and incubated at RT for 1 h with gentle shaking. After the first reaction, the membrane was soaked in T/PBS and washed (the membrane was washed between every reaction) and soaked in T/PBS containing 1:5000 dilution horseradish peroxidase-labelled anti-mouse IgG (Jackson ImmunoResearch; USA). Subsequently, it was incubated at RT for 1 h with gentle shaking. In addition to indirect staining, the membrane

was soaked in a 1:5000 dilution of HRP-labeled MAb, followed by blotting and blocking, when HRP-labeled MAb was used for direct staining. After the final wash, the membrane was soaked in 10 mL of a color developer, Easy West Blue (ATTO), and incubated at RT for 5 min with gentle shaking. After color development, the membrane was washed with water to terminate the reaction. The color intensities of all dotted points were scanned using an image analyzer, and images were captured.

## Serum neutralization test for the detection of anti-PMT dermonecrotic toxin

A microneutralization test was performed using a 96-well microplate to detect the neutralizing antibodies. *P. multocida* strain S70 was cultured in a heart infusion broth for 18 h at 37°C using a shake culture, and the grown cells were concentrated followed by disruption via sonication. The centrifuged supernatant was filtered using a 0.45 μm filter as a challenge Pm toxin. Pig sera were heated at 56°C for 30 min before the assay. The sera and Pm toxin were diluted with supplemented Eagle's minimal essential medium (E-MEM; Nissui Pharmaceutical Co., Ltd.; Tokyo, Japan) containing trypticase soy broth (Becton, Dickinson and Company; Tokyo, Japan), sodium bicarbonate (FUJIFILM Wako Pure Chemical Corporation.; Osaka, Japan), L-glutamine (FUJIFILM Wako Pure Chemical Corporation.; Osaka, Japan), and gentamycin (FUJIFILM Wako Pure Chemical Corporation.; Osaka, Japan). Subsequently, 50 μL of serial duplicate two-fold dilutions was mixed with an equal volume of Pm toxin containing 80 units/mL and incubated at 37°C for 1 h. Thereafter, 100 μL of bovine embryonic lung cells, EBL cell (ACC 192) suspension ($4 \times 10^5$ cells/mL) in E-MEM and Hanks (Nissui Pharmaceutical Co., Ltd.; Tokyo, Japan) mixture medium containing 10% FBS, sodium bicarbonate, L-glutamine, penicillin (Meiji Seika Pharma Co., Ltd.; Tokyo, Japan), streptomycin (Meiji Seika Pharma Co., Ltd.; Tokyo, Japan), and kanamycin (FUJIFILM Wako Pure Chemical Corporation.; Osaka, Japan) were added to the toxin-serum mixtures and cultured at 37°C in 5% $CO_2$ for 7 days. The neutralizing antibody titer was expressed as the reciprocal of the highest serum dilution that inhibited the development of the cytopathic effect (CPE).

## Serum collection from pigs inoculated with AR and AR/ER vaccine for the neutralization test

Ten 4-week-old pigs were vaccinated twice with two lots each of AR and AR/ER vaccines at a 4-week interval. The sera were collected 4 weeks after the last vaccination and used for the neutralization test. The sera of pigs vaccinated with AR vaccine Lot A and AR/ER vaccine Lot C were obtained from a field farm. The sera of pigs vaccinated with AR vaccine Lot B and AR/ER vaccine Lot D were obtained through animal experiments in an in-house animal facility.

## Ethics statement

Twenty 3-month-old LWD crossbred pigs (Landrace, Large White, and Duroc) were used in the in-house animal study and acclimated for 7 days prior to the study. The pigs had free access to water and were fed a commercial diet (Nosan Corporation; Yokohama, Japan). Their health and behavior were monitored daily. The pigs were retained before immunization and blood collection. At the end of the study period, all pigs were pre-anesthetized with pentobarbital sodium and euthanized via exsanguination by dissecting the axillary artery. Since approved and commercially available vaccines were used in this study, we hypothesized that there would be no deaths or serious adverse events during the study period; therefore, no humane endpoints were set. This study was approved by the internal committee of KM Biologics Co., Ltd.,

and conducted in accordance with the "Guidelines for Proper Conduct of Animal Experiments" (Protocol No.: R200401-2, Approval No.: B20-028). All experimenters and staff received annual training. Only registered individuals conducted animal experiments.

## Results

### Adsorption efficiency of PMT on aluminum hydroxide at different mixing ratios

Different concentrations of PMT and aluminum hydroxide were used to prepare PMT-alum in according to the method of commercial vaccine production. Subsequently, the amount of PMT in the final product was analyzed to determine the influence of the mixing ratio of PMT and aluminum hydroxide on the adsorption efficacy.

Different concentrations of PMT (500, 100, and 10 μg/mL) were mixed with an equal volume of 10 mg/mL of aluminum hydroxide to prepare PMT-alum to confirm the effect of PMT concentration on the PMT-alum preparation process. In addition, the PMT-alum prepared using 500 μg/mL PMT was diluted five and 50 times with 5 mg/mL aluminum hydroxide, which were assumed to be 50 and 5 μg/mL of PMT in PMT-alum, respectively. The samples from each preparation were diluted to a concentration of 1:400 and processed for dot blotting and indirect immunostaining. The results are presented in Fig 1 and S1 Table. The dots immobilized with PMT-alum prepared from 500, 100, and 10 μg/mL of starting materials, which may be half the concentration in PMT-alum, showed different intensities according to the PMT concentration (Fig 1A). A similar result was obtained for the dot immunostaining assay of dilutions of the PMT-alum preparation (Fig 1B). No differences were observed between 50 and 5 μg/mL of the starting material PMT-alum (intensity of 10793.9 and 3353.5, respectively), which was prepared by diluting PMT-alum five and 50 times (intensity of 10572.2 and 4529.1, respectively). For the unpaired t-test between the former and latter samples, the *p*-values were 0.81 and 0.18, respectively.

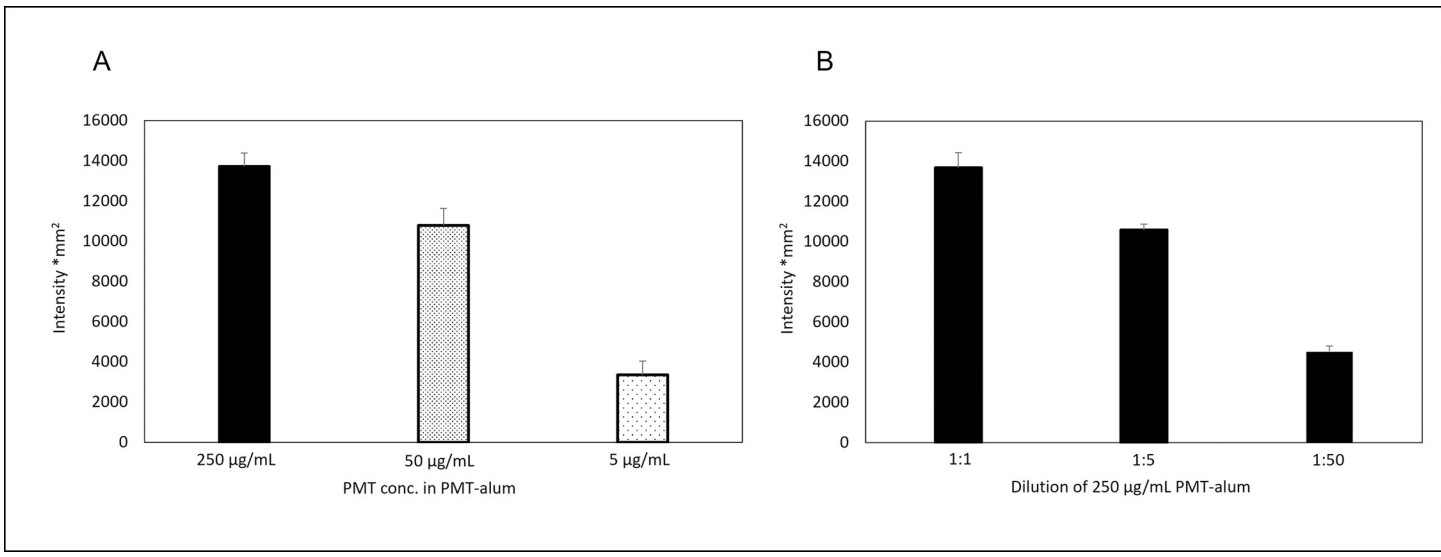

**Fig 1. The intensities of the dot blotting spot of PMT-alum prepared from diluted PMT solutions compared with those of the dilutions of the PMT-alum sample.** A) PMT dilution of 500, 100, and 10 μg/mL was mixed with an equal volume of 10 mg/mL aluminum hydroxide and integrated into PMT-alum. B) PMT concentrations were adjusted to 50 and 5 μg/mL PMT-alum via dilution at 1:5 and 1:50 of PMT-alum sample prepared from 500 μg/mL PMT. The data shows average of intensities (n = 3), and error bars denote SEs. PMT, toxins of *P. multocida;* conc, concentration.

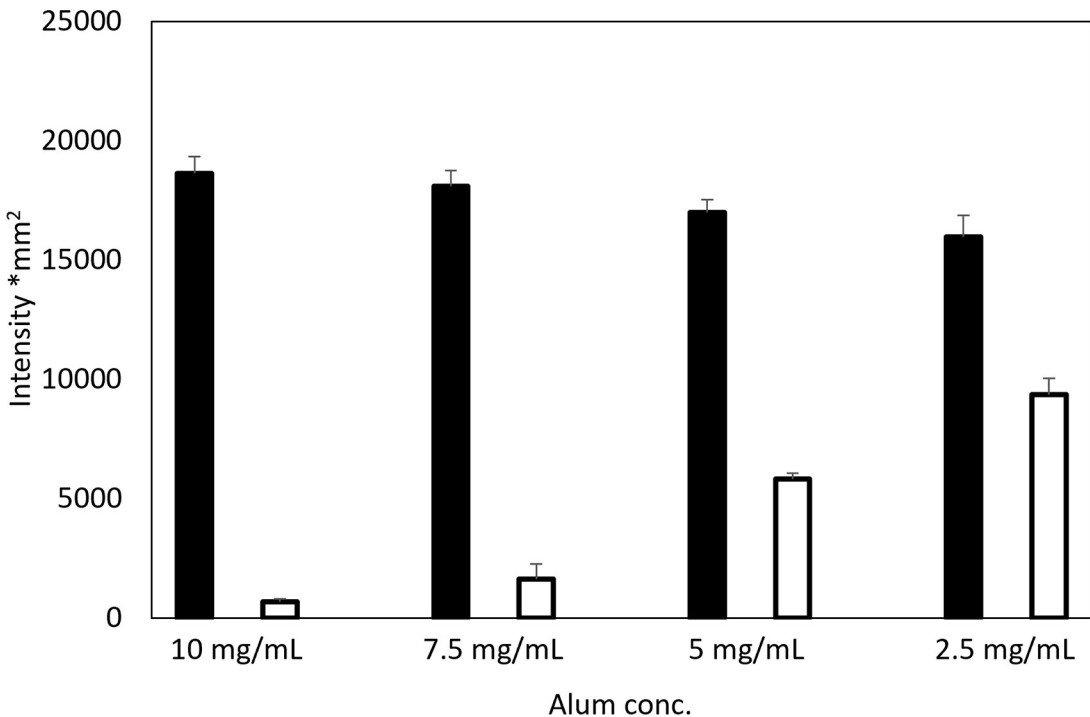

**Fig 2. Intensities of the dot blotting spots of PMT-alum and the supernatants of the mixture before fixation.** A constant concentration of PMT, 500 μg/mL, was mixed with an equal volume of aluminum hydroxide diluted to 10, 7.5, 5, and 2.5 mg/mL and integrated into PMT-alum. Before fixation, the supernatants were collected from the PMT and aluminum hydroxide mixtures. The black bars indicate the average of intensities of the PMT-alum, and the white bars indicate those of the supernatants (n = 3). An error bars denote SEs. PMT, toxins of *P. multocida;* conc, concentration.

Subsequently, different concentrations of aluminum hydroxide (10, 7.5, 5, and 2.5 mg/mL) were mixed with an equal volume of 500 μg/mL PMT to prepare PMT-alum to confirm the effect of the aluminum hydroxide concentration on the process. A portion of the supernatant was collected immediately before fixation to detect non-adsorbed PMT. The PMT-alum and supernatant of the mixtures were diluted to a concentration of 1:1600 and 1:400, respectively, and then processed for dot blotting and immunostaining assays. The results are presented in Fig 2 and S2 Table. The intensities of the PMT-alum samples prepared using different concentrations of aluminum hydroxide varied; however, their levels decreased with the reduction in the aluminum hydroxide concentration. The intensities of the PMT-alum samples prepared from aluminum hydroxide (10, 7.5, 5, and 2.5 mg/mL) were 18639, 18108, 16991, and 15976, respectively. A positive reaction was observed at lower aluminum concentrations in the dot immunostaining assay of the supernatant of the PMT and aluminum hydroxide mixture. The intensities of the supernatant of the mixture at 10, 7.5, 5, 2.5 mg/mL were 578, 1641, 5817, and 9378, respectively.

## Comparison of SDS-PAGE and dot blotting assay

We tried to desorb PMT from aluminum hydroxide; PMT were eluted by high ionic strength (0.3 M phosphate buffer) and concentrated with acetone. The desorbed PMT were quantified by SDS-PAGE. The results are shown in Table 1 and S3 Table. The estimated values of PMT by this method were lower than theoretical values calculated from the original PMT-alum. The recovery rates varied between 8 to 31 percent.

**Table 1. Desorption of PMT from aluminum hydroxide and its recovery rate.**

| Theoretical value (µg/mL) | 200 | 160 | 100 | 80 |
|---|---|---|---|---|
| Determined value (µg/mL) | 63.8 | 49.8 | 11.9 | 6.4 |
| Recovery rate (%) | 31 | 31 | 11 | 8 |

The lowest concentration at which a band was observed was 40 µg/mL PMT-alum, and 80–200 µg/mL PMT-alum were quantifiable (the band intensities of 50, 40 µg/mL PMT-alum were out of range).

The data in Figs 1 and 2 indicate that 10 mg/mL of aluminum hydroxide could adsorb almost all of the 500 µg/mL PMT to form 250 µg/mL PMT-alum. Therefore, standard serial dilution ranging from 200 to 5 µg/mL of PMT-alum was prepared via the dilution of the 250 µg/mL PMT-alum stock. In this experiment, standard dilutions of 200, 160, 100, 80, 50, 40, 25, 20, 10, and 5 µg/mL of PMT-alum were used to create a calibration line. Simultaneously, the same dilutions of PMT standard were prepared. As shown in Fig 3, PMT-alum standard and PMT standard showed almost the same linearity and slope. Based on this, we thought that it would be possible to quantify by dot blot assay using a standard curve drawn with PMT-alum and validated it. The results are shown in Table 2 and S4 Table.

## PMT content and immunogenicity of the AR vaccine product

The abovementioned experiments demonstrated that dot blotting assay could detect and quantify PMT antigens adsorbed on the adjuvant in vaccines. The sera of pigs who had received commercial AR and AR/ER vaccines were collected to evaluate the correlation between the PMT content and antibody titers. Two samples collected from each of the AR and AR/ER vaccine groups were used in the established dot immunoassay. The PMT content was estimated by comparing its intensity with that of the standard line. The sera collected from pigs vaccinated with these four vaccine lots were tested for the presence of neutralizing antibodies against PMT. Table 3 presents the PMT content in the vaccine and the geometric mean antibody titer of every 10 serum samples. The PMT content in the vaccine products of AR Lot A, AR Lot B, AR/ER Lot C, and AR/ER Lot D was approximately 50, 80, 80, and 110 µg of

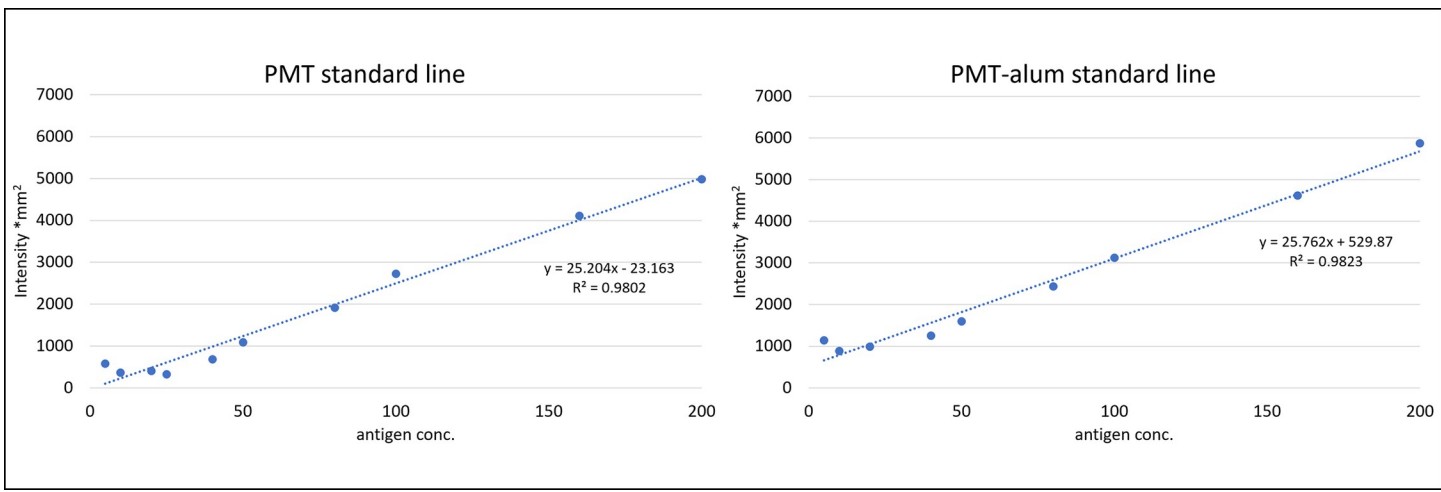

**Fig 3. Serial standard dilution ranging from 200 to 5 µg/mL PMT and PMT-alum was used in the dot immunoassay.** Intensity lines of the PMT (left) and PMT-alum (right). PMT, toxins of *P. multocida;* conc, concentration.

**Table 2. The results of validation.**

| Evaluation items | Result | |
|---|---|---|
| Recovery rate | 125% | |
| Precision | Repeatability | Standard deviation, 9.6 µg/mL<br>Relative standard deviation, 15.3%<br>Confidence interval for standard deviation, 6.2 to 21.2 µg/mL |
| | Intermediate Precision | Standard deviation, 8.1 µg/mL<br>Relative standard deviation, 13.0%<br>Confidence interval for standard deviation, 5.2 to 18.0 µg/mL |
| Limit of quantification | The limit of quantification calculated from the blank measurement value and the slope of the calibration line was estimated to be 0.81 µg/mL | |
| linearity | The multiple correlation coefficient ($R^2$) of the calibration line obtained from the three calibration lines was 0.9763 | |

PMT, respectively, and these vaccines elicited antibodies in pigs at geometric mean titers of 4.0, 16.0, 14.9, and 13.9, respectively. The individual data are shown in S4 and S5 Tables.

## Discussion

Swine AR is caused by PMT [6], which induces the production of neutralizing antibodies in the host and alleviates the disease symptoms [1]. Thus, PMT is an essential component of AR vaccines for pigs, and these vaccines contain adequate concentrations of antigens for antibody induction. In principle, the efficacy of the vaccine, that is, its ability to induce antibodies, is correlated with the antigen dose. Antigen doses play a significant role in inactivated vaccines as they have no replication activity and lower prolonged antigen persistence than live vaccines [10]. Antigen concentration in a vaccine must be measured to evaluate the effects of inactivated vaccines and toxoids. However, as a method for quantifying the antigen bound to the aluminum adjuvant is yet to be established, the antigen concentration in the final product is calculated based on the values measured during this process. Animal experiments are often used as an alternative approach for quantifying the antigen concentration in the final product to determine efficacy.

  We tried to elute PMT from PMT-alum using a high phosphate ion extraction method and quantify by SDS-PAGE. Although quantification was possible, the extraction efficiency was low and variable. Although this elution method is mild that does not affect the antigenicity, but it was found that accurate quantification by this method would be difficult from the viewpoint of the extraction efficiency. AR and AR/ER vaccines are combined vaccines containing multiple antigens. We consider it is difficult to find for the optimal antigen extraction method for all antigens. A dot blotting and immunostaining method was established to measure the concentration of PMT antigens bound to the aluminum hydroxide adjuvant in the present study. Quantification by dot blotting is a method for immunological quantification of PMT without

**Table 3. The PMT content of AR and AR/ER vaccines and the neutralization antibody titer elicited by these vaccines.**

| | AR Lot A | AR Lot B | AR/ER Lot C | AR/ER Lot D |
|---|---|---|---|---|
| **PMT content (µg/mL)** | 45.2 | 77.5 | 84.8 | 107.0 |
| **Antibody titer (GMT, n = 10)** | 4.0 | 16.0 | 14.9 | 13.9 |

PMT content showed the average of the value measured three times

AR, atrophic rhinitis; PMT, toxins of P. multocida; GMT, geometric mean titers

desorption from aluminum hydroxide, and it is useful as a method for quantifying antigens in vaccines because it is easier to operate than extraction methods. This method could quantify PMT in vaccine products with high specificity and sensitivity. Since the MAb used in this study can neutralize the dermonecrotic PMT, the antigen detected using this method is the neutralizing epitope of PMT. Therefore, the amount of antigen quantified using this method is equivalent to the concentration of PMT. This method can also detect antigens bound to aluminum hydroxide, indicating that AR vaccines can appropriately induce neutralizing antibodies against PMT. MAb did not react with the other antigens in the vaccine, such as the dermonecrotic toxin of *B. bronchiseptica* and the surface-protective antigen type A of *E. rhusiopathiae* (S1 Fig).

Aluminum-containing adjuvants are widely used in commercial vaccines due to their safety and enhanced potency. Aluminum hydroxide has an isoelectric point of approximately 11 and adsorbs acidic proteins due to the presence of attractive electrostatic forces [11]. Ganfield et al. [12] reported that the protective antigens of *P. multocida* have precipitinogenic activity in the pH range of 3–4, with the activity peaking at a pH of 3.7. Therefore, PMT appeared to bind effectively to aluminum hydroxide. In the current vaccine production method, the antigen concentration measured by SDS-PAGE is converted to a concentration after binding with aluminum hydroxide. This concentration is calculated by assuming that the antigen was adsorbed 100% by aluminum hydroxide; however, the adsorption efficiency was not confirmed. Therefore, various antigen and aluminum hydroxide concentrations were prepared, and their adsorption efficiencies were evaluated. The experimental simulation revealed that PMT could be detected in the supernatant as an inverse proportion of the concentration of aluminum hydroxide by assigning different concentrations of aluminum hydroxide and evaluating the binding ability with 500 μg/mL PMT. Indeed, PMT-alum prepared under these conditions shows the same dot blotting response as PMT alone. Since the concentration of PMT obtained during manufacturing is usually approximately 500 μg/mL, it was assumed that all antigens can be adsorbed by 10 mg/mL aluminum hydroxide. However, if the yield of PMT is exceedingly high, the actual amount of antigen in the product may be less than the value converted from the PMT concentration measured by SDS-PAGE, as the adsorption capacity of aluminum hydroxide will be exceeded (PMT will be saturated). It would be possible to closely monitor the concentration of the PMT antigen in the final product using this method, which could not be measured previously. In this study, the neutralizing antibodies elicited by the vaccines were correlated with the amount of PMT antigen in the vaccine products (Table 3), indicating that this in vitro method reflected the in vivo test results.

In conclusion, this method can be a valuable tool that can be used for quality control of vaccine consistency during production. For this vaccine, the minimum effective antigen amount of PMT is 10 μg/mL, and the vaccine is typically adjusted to contain at least four times that amount with a margin. This amount of PMT is well within the quantification range of the method presented here. This quantification method can be easily applied to other antigens if suitable antibodies are available. Furthermore, there is an opportunity to reduce the number of test animals, such as mice, used in quality control examinations by replacing animal models, which is beneficial from an animal welfare perspective. Further tests such as evaluating the storage condition and ensuring lot-to-lot consistency of the standard samples are required to establish this method as a quality test in the future.

## Supporting information

**S1 Table. The raw data of Fig 1.**
(DOCX)

**S2 Table. The raw data of Fig 2.**
(DOCX)

**S3 Table. The raw data of Table 1.**
(DOCX)

**S4 Table. The raw data of Tables 2 and 3.** The validation analysis for Table 2 is included.
(DOCX)

**S5 Table. The raw data of Table 3.** Individual neutralizing antibody titer following vaccination is described.
(DOCX)

**S1 Fig. Confirmation of MAb specificity.** The MAb, which used in this study, did not react with the other antigens in the vaccine.
(DOCX)

## Acknowledgments

The authors are grateful to Prof. Yasuhiko Horiguchi for providing hybridomas. The authors also thank Dr. Tomoyuki Tsuda for their assistance in preparing this manuscript. We would like to thank Honyaku Center Inc. (www.honyakuctren.com) and Editage (www.editage.com) for English language editing.

## Author Contributions

**Data curation:** Masaya Yoshimura, Kiyotaka Suenaga.

**Formal analysis:** Masaya Yoshimura, Youko Honda, Emi Yonemitsu, Kasumi Kuraoka.

**Methodology:** Masaya Yoshimura, Youko Honda, Kiyotaka Suenaga.

**Project administration:** Kiyotaka Suenaga.

**Supervision:** Kiyotaka Suenaga.

**Visualization:** Masaya Yoshimura.

**Writing – original draft:** Masaya Yoshimura.

**Writing – review & editing:** Masaya Yoshimura, Kiyotaka Suenaga.

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
