## [Decision Letter · Decision Letter 0]

20 Jul 2023

PONE-D-23-12736Method for quantifying the Pasteurella multocida antigen adsorbed on aluminum hydroxide adjuvant in swine atrophic rhinitis vaccinePLOS ONE

Dear Dr. Yoshimura,

Thank you for submitting your manuscript to PLOS ONE. After careful consideration, we feel that it has merit but does not fully meet PLOS ONE’s publication criteria as it currently stands. Therefore, we invite you to submit a revised version of the manuscript that addresses the points raised during the review process.

1) Provide experimental data showing how much (antigen) is bound in alum;2) Provide a rational for the inactivation process after adsorption of the antigen to alum. This is not usual. In general, the antigen is treated with formaldehyde and then adsorbed on alum;3) Please, answer to all the comments raised by both reviewers.

We look forward to receiving your revised manuscript.

Kind regards,

Paulo Lee Ho, Ph.D.

Academic Editor

PLOS ONE

Journal Requirements:

Reviewers' comments:

Reviewer's Responses to Questions

**Comments to the Author**

1. Is the manuscript technically sound, and do the data support the conclusions?

Reviewer #1: Partly

Reviewer #2: Partly

2. Has the statistical analysis been performed appropriately and rigorously? 

Reviewer #1: I Don't Know

Reviewer #2: No

3. Have the authors made all data underlying the findings in their manuscript fully available?

Reviewer #1: Yes

Reviewer #2: Yes

4. Is the manuscript presented in an intelligible fashion and written in standard English?

Reviewer #1: Yes

Reviewer #2: Yes

5. Review Comments to the Author

Reviewer #1: I am concerned about how the toxin was bound to the adjuvant since Phosphate interferes and was included in all mixing and testing buffers. Because of the including of the phosphate buffer I don't agree with the assumption of 100% bound. No measurements were made to evaluate the amount of toxin bound and no evaluation was done to look at parameters to increase binding (time and buffer). This is a significant gap to the work conducted.

Reviewer #2: Yoshimura and colleagues describe the in vitro quantification of alum-adsorbed P. multocida toxin (PMT). In principle, the manuscript addresses an important issue and the experiments follow a clear rationale. However, the authors should demonstrate the quantitative desorption of antigens. It remains unclear, how much antigen is still retained with the alum. In addition, it seems that all experiments have been performed once and without replicates. So, there is no statistical evaluation of the data. The broad body of literature on antigen desorption should be considered and appropriately cited.

Specific comments:

Line 86 ff. – adsorption of PMT on aluminium hydroxide

- The process of alum adsorption is relatively fast, but the duration of the adsorption process has an influence on the strength of binding. An over night adsorption might not be representative for commercial products.

- Usually, antigens -such as tetanus toxoids- are formalin treated before alum adsorption. What is the rationale for the fixation step after the adsorption?

Line 101 – Neutralizing activity of anti PMT MAb: Please provide a reference.

Line 106 – Dot blotting

- Desorption of antigens from aluminum hydroxide salts is achieved by high ionic strength and/or high pH of the desorption buffer (as alum reaches neutral charge at ~pH 9). In addition, the process is time dependent. The process described in the manuscript uses neither of these mechanisms. It is hard to conceive that quantitative desorption was reached. This should be demonstrated, also with commercial vaccine batches.

Line 188 – Dot blotting Results Fig 1 & 2

- Desorption should be performed in replicates. Results should be statistically analyzed.

Line 202 ff. – Adsorption using different concentrations of Alum

- It is surprising to see non-adsorbed antigen with the highest alum concentration. Is there an explanation, why the antigen content decreases from 10 to 7.5 mg/ml Alum, although the amount of free antigen seems to be lowest with 7.5 mg/ml?

Line 228 – You probably refer to “raw” data?

Line 248 ff. – The PMT content of AR and AR/ER vaccines

- Is there data available, whether other antigens, such as ER, influence the detection of PMT?

- What is the 95% confidence interval of the antibody titres? Are the titres significantly different?

6. PLOS authors have the option to publish the peer review history of their article (what does this mean?). If published, this will include your full peer review and any attached files.

Reviewer #1: **Yes: **Jessica White

Reviewer #2: No

---

## [Author Response · Author response to Decision Letter 0]

7 Sep 2023

Thank you for your review and comments on our paper PONE-D-23-12736.

We have carefully considered the comments and have revised the manuscript accordingly. 

Our responses are provided point-by-point manner in the file labeled 'Response to Reviewers'.

We hope that the revised version is now suitable for publication in PLOS ONE.

---

## [Decision Letter · Decision Letter 1]

12 Oct 2023

PONE-D-23-12736R1Method for quantifying the Pasteurella multocida antigen adsorbed on aluminum hydroxide adjuvant in swine atrophic rhinitis vaccinePLOS ONE

Dear Dr. Yoshimura,

Thank you for submitting your manuscript to PLOS ONE. After careful consideration, we feel that it has merit but does not fully meet PLOS ONE’s publication criteria as it currently stands. Therefore, we invite you to submit a revised version of the manuscript that addresses the points raised during the review process.

1) The authors should provide statistical analysis as requested;2) The authors should consider revising the manuscript and provide the determination of the antigen adsorbed on alumn;3) Please see the comments raised by the reviewers.

We look forward to receiving your revised manuscript.

Kind regards,

Paulo Lee Ho, Ph.D.

Academic Editor

PLOS ONE

Reviewers' comments:

Reviewer's Responses to Questions

**Comments to the Author**

1. If the authors have adequately addressed your comments raised in a previous round of review and you feel that this manuscript is now acceptable for publication, you may indicate that here to bypass the “Comments to the Author” section, enter your conflict of interest statement in the “Confidential to Editor” section, and submit your "Accept" recommendation.

Reviewer #1: All comments have been addressed

Reviewer #2: (No Response)

2. Is the manuscript technically sound, and do the data support the conclusions?

Reviewer #1: Yes

Reviewer #2: No

3. Has the statistical analysis been performed appropriately and rigorously? 

Reviewer #1: Yes

Reviewer #2: No

4. Have the authors made all data underlying the findings in their manuscript fully available?

Reviewer #1: Yes

Reviewer #2: Yes

5. Is the manuscript presented in an intelligible fashion and written in standard English?

Reviewer #1: Yes

Reviewer #2: Yes

6. Review Comments to the Author

Reviewer #1: Thank you for your careful review and response to the questions. Repeat testing has been conducted and revisions have been made to address the concerns of the reviewers.

Reviewer #2: The authors have not sufficiently addressed the reviewers concerns: Antigen that is adsorbed to the alum matrix is not accessible to immune detection by dot blotting, but is immunologically active. If the assay is meant to measure the antigen content IT IS NECESSARY to quantitatively determine the amount of antigen. If the authors refuse to provide evidence for this, the paper is not suited for publication. Why has the comment regarding the broad body of literature been ignored? Please also see the detailed Reviewers Comments as attached.

7. PLOS authors have the option to publish the peer review history of their article (what does this mean?). If published, this will include your full peer review and any attached files.

Reviewer #1: No

Reviewer #2: No

---

## [Author Response · Author response to Decision Letter 1]

13 Nov 2023

Thank you for your reviewing of our revised version. We have carefully read the comments of reviewer 2, which were the basis for the decision of "major revision".

The main purpose of this paper is to attempt to directly quantify antigens in existing commercial vaccines using by dot immunoassay. However, reviewer 2 states that "antigen that adsorbed to the alum matrix is not accessible to immune detection by dot blotting", but this is a major misunderstanding of what our paper claims. In addition, the some comments are "suggestions on the manufacturing method", but this point is a different issue from the purpose of this paper. It is not easy to change a process that has already been approved, and that's not what we want to do. Reviewer 1 seemed to agree with our previous response that the antigens used in this quantitative system are prepared in accordance with the veterinary drug approvals.

We would like you, the Academic Editor, to review the gist of this paper and decide if corrections are needed in response to Reviewer 2's points.

---

## [Decision Letter · Decision Letter 2]

20 Dec 2023

PONE-D-23-12736R2Method for quantifying the Pasteurella multocida antigen adsorbed on aluminum hydroxide adjuvant in swine atrophic rhinitis vaccinePLOS ONE

Dear Dr. Yoshimura,

Thank you for submitting your manuscript to PLOS ONE. After careful consideration, we feel that it has merit but does not fully meet PLOS ONE’s publication criteria as it currently stands. Therefore, we invite you to submit a revised version of the manuscript that addresses the points raised during the review process.

We look forward to receiving your revised manuscript.

Kind regards,

Paulo Lee Ho, Ph.D.

Academic Editor

PLOS ONE

Reviewers' comments:

Reviewer's Responses to Questions

**Comments to the Author**

1. If the authors have adequately addressed your comments raised in a previous round of review and you feel that this manuscript is now acceptable for publication, you may indicate that here to bypass the “Comments to the Author” section, enter your conflict of interest statement in the “Confidential to Editor” section, and submit your "Accept" recommendation.

Reviewer #3: (No Response)

2. Is the manuscript technically sound, and do the data support the conclusions?

Reviewer #3: Partly

3. Has the statistical analysis been performed appropriately and rigorously? 

Reviewer #3: N/A

4. Have the authors made all data underlying the findings in their manuscript fully available?

Reviewer #3: Yes

5. Is the manuscript presented in an intelligible fashion and written in standard English?

Reviewer #3: Yes

6. Review Comments to the Author

Reviewer #3: The manuscript is a good attempt. The desorption of antigens from Aluminum hydroxide adjuvant without disturbing the antigenicity for ELISA methods is always a challenge. The authors report the use of dot blot assay for quantification of antigens. Dot blot assay is a semiquantitative method. The authors report its use for quantitative use for gel adsorbed vaccines. The data is good; however, needs a very convincing and robust discussion in the manuscript on following.

1. The data should be compared to method in prior art which is quantitative in function

2. The data set should also include robust method validation data as per ICH guidelines on quantitative methods with focus on recovery and precision

3. The manuscript should include a detailed development account on how desorption conditions were designed to get a good recovery.

4. Dot blot method LOQ and how it will be used in vaccine testing should be stressed upon in discussion.

7. PLOS authors have the option to publish the peer review history of their article (what does this mean?). If published, this will include your full peer review and any attached files.

Reviewer #3: **Yes: **Sunil Gairola

---

## [Author Response · Author response to Decision Letter 2]

13 Feb 2024

Thank you for reviewing our paper. We have carefully corrected the points you commented on. We hope that this revised version will be acceptable for publication.

---

## [Decision Letter · Decision Letter 3]

21 Mar 2024

Method for quantifying the Pasteurella multocida antigen adsorbed on aluminum hydroxide adjuvant in swine atrophic rhinitis vaccine

PONE-D-23-12736R3

Dear Dr. Yoshimura,

We’re pleased to inform you that your manuscript has been judged scientifically suitable for publication and will be formally accepted for publication once it meets all outstanding technical requirements.

Kind regards,

Paulo Lee Ho, Ph.D.

Academic Editor

PLOS ONE

Additional Editor Comments (optional):

Reviewers' comments:

Reviewer's Responses to Questions

**Comments to the Author**

1. If the authors have adequately addressed your comments raised in a previous round of review and you feel that this manuscript is now acceptable for publication, you may indicate that here to bypass the “Comments to the Author” section, enter your conflict of interest statement in the “Confidential to Editor” section, and submit your "Accept" recommendation.

Reviewer #4: All comments have been addressed

2. Is the manuscript technically sound, and do the data support the conclusions?

Reviewer #4: Yes

3. Has the statistical analysis been performed appropriately and rigorously? 

Reviewer #4: Yes

4. Have the authors made all data underlying the findings in their manuscript fully available?

Reviewer #4: Yes

5. Is the manuscript presented in an intelligible fashion and written in standard English?

Reviewer #4: Yes

6. Review Comments to the Author

Reviewer #4: Measuring the concentration of antigen absorbed to aluminum adjuvant is difficult technical aspiration of this manuscript. The authors successfully accomplished this goal for this important vaccine. Unfortunately, there was only a qualitative relationship between the measured vaccine antigen concentration and the in vivo antibody titers. This not unexpected however, because of the high variability of the immune response in animals.

7. PLOS authors have the option to publish the peer review history of their article (what does this mean?). If published, this will include your full peer review and any attached files.

Reviewer #4: **Yes: **Patrick Ahl

---

## [Editor Report · Acceptance letter]

7 May 2024

PONE-D-23-12736R3 

PLOS ONE

Dear Dr. Yoshimura, 

I'm pleased to inform you that your manuscript has been deemed suitable for publication in PLOS ONE. Congratulations! Your manuscript is now being handed over to our production team.

Kind regards, 

on behalf of

Dr. Paulo Lee Ho 

Academic Editor

PLOS ONE